# GABA-Positive Astrogliosis in Sleep-Promoting Areas Associated with Sleep Disturbance in 5XFAD Mice

**DOI:** 10.3390/ijms24119695

**Published:** 2023-06-02

**Authors:** Victor James Drew, Mincheol Park, Tae Kim

**Affiliations:** Department of Biomedical Science and Engineering, Gwangju Institute of Science and Technology, Gwangju 61005, Republic of Korea; victor.j.drew@gist.ac.kr (V.J.D.); eyejor@gist.ac.kr (M.P.)

**Keywords:** Alzheimer’s disease, 5XFAD, sleep disturbance, reactive astrogliosis, GABA

## Abstract

Sleep disturbances, a debilitating symptom of Alzheimer’s disease (AD), are associated with neuropathological changes. However, the relationship between these disturbances and regional neuron and astrocyte pathologies remains unclear. This study examined whether sleep disturbances in AD result from pathological changes in sleep-promoting brain areas. Male 5XFAD mice underwent electroencephalography (EEG) recordings at 3, 6, and 10 months, followed by an immunohistochemical analysis of three brain regions associated with sleep promotion. The findings showed that 5XFAD mice demonstrated reduced duration and bout counts of nonrapid eye movement (NREM) sleep by 6 months and reduced duration and bout counts of rapid eye movement (REM) sleep by 10 months. Additionally, peak theta EEG power frequency during REM sleep decreased by 10 months. Sleep disturbances correlated with the total number of GFAP-positive astrocytes and the ratio of GFAP- and GABA-positive astrocytes across all three sleep-associated regions corresponding to their roles in sleep promotion. The presence of GABRD in sleep-promoting neurons indicated their susceptibility to inhibition by extrasynaptic GABA. This study reveals that neurotoxic reactive astrogliosis in NREM and REM sleep-promoting areas is linked to sleep disturbances in 5XFAD mice, which suggests a potential target for the treatment of sleep disorders in AD.

## 1. Introduction

Disturbances of the normal sleep cycle have been reported in roughly one-third of Alzheimer’s disease (AD) patients [1], which results in daytime drowsiness, nighttime awakenings, and other impairments [2,3,4,5]. AD, the most common cause of dementia [6,7], is believed to result from the accumulation of amyloid beta (Aβ) plaques and tau neurofibrillary tangles (NFTs) [8,9,10]. In AD, the progression of Aβ distribution does not occur uniformly throughout the brain. Human AD studies have demonstrated that the Aβ pathology initiates in regions such as the medial orbitofrontal cortex, precuneus, hippocampus, and cingulate cortices before advancing into subcortical regions, including the amygdala, hypothalamus, and thalamus, before ultimately reaching the brainstem and cerebellum at the latest stages [11,12]. Many symptoms of AD can be traced to amyloidosis or tauopathies in brain regions that control the specific functions being affected. Examples include the association between memory loss in AD and amyloid and tau accumulation in the cortex and hippocampus [13,14], and intraneuronal Aβ aggregation in the amygdala, which enhances fear and anxiety in AD transgenic mice [15].

Sleep and wakefulness are controlled by complex systems of neurocircuitry and glial support. Among these are various sleep-promoting neuronal groups, including neuronal nitric oxide synthase (nNOS)-containing neurons of the cerebral cortex and galaninergic neurons of the ventrolateral preoptic area (VLPO), both of which promote nonrapid eye movement (NREM) sleep duration and homeostatic sleep drive [16,17], and the rapid eye movement (REM) sleep-promoting cholinergic neurons of the laterodorsal tegmentum (LDT) [18].

Astrocytes are recognized for performing several critical functions in the brain, including waste removal, maintaining interstitial homeostasis, structural scaffolding, and forming glial scar tissue for injury repair [19]. Astrocytes have been shown to modulate sleep pressure accumulation and the associated cognitive consequences via a pathway involving A1 receptors [20]. Furthermore, after exposure to insult or injury, astrocytes undergo morphological, chemical, and functional changes in response to the pathological situation [21]. The transition into this new state is referred to as “reactive astrogliosis” [21]. One functional change in reactive astrocytes observed in animal models of AD involves the degradation of Aβ and biosynthesis of GABA, followed by the extracellular release of GABA, which can potentially suppress neighboring cells possessing extrasynaptic GABA receptors [22].

Taken together, multiple cellular and molecular factors are likely involved in the sleep disturbances observed in AD. However, the pathogenic mechanism of sleep disturbances in AD remains unknown. Therefore, we hypothesize that sleep is disturbed by pathological changes in sleep-promoting brain regions in AD and sought to investigate the association between sleep disturbances and the progression of the pathology in the cortex, VLPO, and LDT in a mouse model of AD.

## 2. Results

### 2.1. Progressive Sleep Disturbances with Age in 5XFAD Mice

The sleep durations of the control and 5XFAD mice were measured using EEG/EMG for a 24 h recording period at 3, 6, and 10 months of age. When comparing genotypes, the 3-month-old 5XFAD mice showed no difference from the control mice in the time spent in wakefulness, NREM sleep, REM sleep, and total sleep. An hour-by-hour sleep analysis revealed similarities in the overall sleep patterns (Figure 1A–C). When compared with the control mice at 6 months of age, the 5XFAD mice showed an increase in wake duration (43.2% vs. 49.7%, *p* < 0.05) and declines in NREM (49.5% vs. 43.2%, *p* < 0.05) and total sleep time (56.8% vs. 50.3%, *p* < 0.05; Figure 1B,C,F). At 10 months of age, a further increase in the wakefulness duration (46.7% vs. 63.7%, *p* < 0.001), a further reduction in NREM sleep (46.7% vs. 32.1%, *p* < 0.001) and total sleep duration (53.3% vs. 36.3%, *p* < 0.001), and an initial decrease in REM sleep duration (6.6% vs. 4.2%, *p* < 0.01; Figure 1C–G) were observed in the 5XFAD mice compared to the control mice. The ratio of REM to total sleep was similar between both genotypes at all ages examined (Figure 1H). The differences in sleep duration between genotypes were more prominent during the dark (active) phase (Figure 1A–C and Appendix A).

Regarding the longitudinal analyses, the control mice showed no differences among all age groups in the durations of vigilance states except in the REM/TS (REM sleep/total sleep) ratio (16.0% at 3 months and 11.3% at 10 months, *p* < 0.01; Figure 1H). In contrast, the 5XFAD mice exhibited changes in the vigilance state durations across age groups. At 6 months, the sleep and wake durations did not differ from the 3-month-old 5XFAD mice. However, decreased sleep durations and increased wakefulness in the 5XFAD mice were significant by 10 months (Figure 1D–G). In addition, the 5XFAD and control mice showed a similar progressive decline in the REM/TS ratio (Figure 1H).

### 2.2. Reduced Numbers of NREM and REM Sleep Episodes in Older 5XFAD Mice

Bout lengths and counts between genotypes and among ages were assessed for evidence of sleep fragmentation. The 5XFAD mice demonstrated a progressive lengthening of wake bouts with age compared to the control mice, whereas the NREM bout lengths remained unchanged for both genotypes across all age groups (Figure 2A,B). Interestingly, the 5XFAD mice demonstrated increased REM bout lengths at 3 and 10 months old compared to the control mice despite neither genotype showing change with age (Figure 2C).

The 5XFAD mice demonstrated a consolidation of episodes for sleep and wakefulness over time. The bout counts of all vigilance states declined with age for the 5XFAD mice compared to the control mice (Figure 2D–F). The wake and NREM bout counts were similar between the 5XFAD and control mice at 3 months but were lower in the 5XFAD mice than in the control mice at 6 and 10 months. The REM bout counts decreased progressively with age in the 5XFAD mice. However, the REM bout counts in the 5XFAD mice did not differ statistically from the control mice until 10 months (*p* < 0.01; Figure 2F). The bout counts remained consistent for the control mice throughout all the vigilance states, except for a temporary increase in the wake bout counts at 6 months. The decrease in the bout counts with age in the 5XFAD mice was more prominent during the dark period than during the light period (Appendix A).

### 2.3. Altered NREM and REM EEG in 5XFAD Mice with Age

A state-dependent analysis of the raw and normalized EEG power spectra was conducted to assess the potential changes in the qualitative features of sleep (Figure 3A–C and Appendix A). Due to their potential implications on NREM and REM sleep efficiency, we examined changes in normalized power between genotypes and among all three age groups at delta (0.5–4 Hz) and theta (5.5–8.5 Hz) frequency bands, respectively (Figure 3D,E). At 3 months, the 5XFAD mice exhibited lower NREM delta power than the control mice (*p* < 0.05; Figure 3D). However, this difference was lost by 6 months as the NREM delta power in the 5XFAD mice increased with age (*p* < 0.001; Figure 3D). The control mice also demonstrated a slight increase in NREM delta power over time, but the change was smaller than in the 5XFAD mice (*p* < 0.05). At 6 and 10 months, the 5XFAD and control mice showed a comparable NREM delta power.

The 5XFAD mice demonstrated a slight decrease in the REM theta power from 3 to 10 months (*p* < 0.05; Figure 3E). However, the decrease did not result in a statistical difference between the control and 5XFAD mice, except at 6 months when the 5XFAD mice showed a temporarily elevated REM theta power (Figure 3E). The control mice did not demonstrate a permanent change in the REM theta power over time. The largest qualitative change in sleep observed in the 5XFAD mice was the reduction in the REM sleep theta peak frequency. Although both the control and 5XFAD mice demonstrated a slowing of the REM sleep theta peak frequency from 6 months to 10 months, the rate of decline was greater in the 5XFAD mice (*p* < 0.01 for controls, *p* < 0.001 for 5XFAD; Figure 3F). This difference contributed to a decrease in the REM sleep theta peak frequency in the 5XFAD mice compare to the control mice at 10 months (*p* < 0.05; Figure 3F).

### 2.4. Progressive Reactive Astrogliosis in the Cortex and VLPO Associated with Decreased NREM Sleep in 5XFAD Mice

The immunohistochemistry of the deep layers of the cerebral cortex and the VLPO of the 5XFAD mice at 3, 6, and 10 months revealed a progressive amyloid pathology and elevated neurotoxic reactive astrogliosis (Figure 4A–G and Figure 5A–G). An age-dependent trend of increasing GABA-positive astrogliosis occurred in proximity to the cortical nNOS neurons and galaninergic neurons in the VLPO (Figure 4B and Figure 5B). Orthogonal plane images confirmed the expression of GABRD in the nNOS neurons and galaninergic VLPO neurons of the 5XFAD and control mice (Figure 4C and Figure 5C), which suggests their susceptibility to extracellular GABA. The 5XFAD mice showed a nearly linear amyloid plaque accumulation in the cerebral cortex between 3 and 10 months, with an increase from 3 to 6 months (*p* < 0.001) and a further increase from 6 months to 10 months (*p* < 0.001; Figure 4E). Plaque accumulation in the VLPO rose more steadily over time with an increase from 3 to 10 months (*p* < 0.01; Figure 5E). An elevated plaque load and mean GFAP intensity were also observed in the cortex, VLPO, and hippocampus (Appendix A).

The 5XFAD mice demonstrated higher quantities of astrocytes by 6 months (*p* < 0.001) and even greater quantities by 10 months compared to the controls (*p* < 0.001; Figure 4F), whereas the VLPO did not show higher quantities of astrocytes compared to the control mice until 10 months (*p* < 0.001; Figure 5F). However, the proportion of GABA-positive astrocytes in the cortex and VLPO both increased at similar rates in the 6-month and 10-month-old 5XFAD mice compared to the age-matched controls (Figure 4G and Figure 5G). Age had no impact on the cortical and VLPO total astrocyte counts or the proportion of GABA-positive astrocytes in the control mice. Interestingly, the cortical nNOS neuron counts and galaninergic neuron counts remained unchanged among both the control and 5XFAD mice across all age groups (Figure 4H and Figure 5H), which suggests that any changes in NREM sleep characteristics were unrelated to a loss of cortical nNOS neurons or galaninergic VLPO neurons.

Pooled data of the control and 5XFAD mice of all age groups were used to assess the relationships between GABA-positive astrocytes in the deep layers of the cortex or the VLPO and various parameters of NREM sleep. The proportion of cortical GABA-positive astrocytes was negatively correlated with NREM sleep duration (r = −0.597, *p* < 0.001; Figure 4I) and NREM bout counts (r = −0.685, *p* < 0.001; Figure 4K) and was positively correlated with NREM bout lengths (r = 0.530, *p* = 0.001; Figure 4J). Similarly, GABA-positive astrogliosis in the VLPO was negatively correlated with NREM sleep duration (r = −0.519, *p* < 0.001; Figure 5I) and NREM bout counts (r = −0.681, *p* < 0.001; Figure 5K) and was positively correlated with NREM bout lengths (r = 0.618, *p* < 0.001; Figure 5J).

### 2.5. Progressive Reactive Astrogliosis at the LDT Associated with REM Sleep Disturbance in 5XFAD Mice

The immunohistochemistry of the LDT in the 5XFAD mice at 3, 6, and 10 months revealed a progressive amyloid pathology and elevated neurotoxic reactive astrogliosis (Figure 6A–G). An age-dependent increased presence of GABA-positive reactive astrogliosis occurred in proximity to the cholinergic neurons in the LDT (Figure 6B). Staining with GABRD confirmed extrasynaptic GABA_A_ receptor expression in the cholinergic neurons in the LDT of the control and transgenic 5XFAD mice at 10 months (Figure 6C), which suggests the susceptibility of these cholinergic neurons to inhibition by extracellular GABA. The 5XFAD mice did not demonstrate an increase in amyloid plaque accumulation in the LDT until 10 months when compared to their levels at 3 months (*p* < 0.01; Figure 6E), whereas the astrocyte counts rose as early as 6 months (*p* < 0.001; Figure 6F). Increases in the plaque load and mean GFAP intensity were also observed in the VLPO (Appendix A). Age had no impact on LDT astrocyte counts in the control mice. (Figure 6F). GABA-positive astrocyte counts in the LDT did not increase until 10 months in the 5XFAD mice compared to age-matched controls (*p* < 0.001; Figure 6G). The quantity of cholinergic neurons in the LDT remained unchanged among both the control and 5XFAD mice of all age groups (Figure 6H), which suggests that any changes in REM sleep characteristics were unrelated to a loss of cholinergic LDT neurons.

Pooled data of the control and 5XFAD mice of all age groups used to assess the relationships between the proportions of GABA-positive astrocytes in the LDT and various parameters of REM sleep showed that the proportion of GABA-positive astrocytes was negatively correlated with REM duration (r = −0.486, *p* = 0.003; Figure 6I) and REM bout counts (r = −0.556, *p* < 0.001; Figure 6K) and was positively correlated with REM bout lengths (r = 0.389, *p* = 0.021; Figure 6J). The proportion of GABA-positive astrocytes was not correlated with the REM theta power (r = 0.218, *p* = 0.209; Figure 6L). However, the proportion of GABA-positive astrocytes was negatively correlated with the theta band peak frequency during REM sleep (r = −0.526, *p* = 0.001; Figure 6M). An additional correlation analysis between the NREM and REM sleep parameters and total astrocyte counts at sleep-promoting regions was conducted and are provided in Appendix A. Enhanced magnification histology images and the GABA intensity in the GFAP area of the cortex, VLPO, and LDT of the control and 5XFAD mice at 10 months are provided in Appendix A.

## 3. Discussion

This study examined the changes in the sleep behavior of a 5XFAD mouse model with respect to the progression of amyloidosis and astrogliosis. An EEG analysis of 24 h recordings revealed decreased NREM and REM sleep durations and bout counts, as well as an altered power spectra, including reduced theta peak frequencies during REM sleep with age. In addition to amyloid plaque accumulation, the 5XFAD mice demonstrated progressively increasing GABA-positive proportions of astrocytes in sleep-promoting regions of the brain. The neurons associated with sleep promotion, including cortical nNOS neurons, galaninergic neurons in the VLPO, and cholinergic neurons in the LDT, were shown to possess extrasynaptic GABA_A_ receptors. Proportions of GABA-positive astrocytes at sleep-promoting regions correlated with various parameters of sleep disturbance in the 5XFAD mice. To our knowledge, this is the first study to propose a potential role of astrocytic GABA in the disturbance of sleep in AD.

An analysis of the sleep–wake patterns in the 5XFAD mice revealed a gradual decline in various sleep parameters with age. The 5XFAD mice showed a similar sleep pattern as the control mice at 3 months. However, by 6 months, the 5XFAD mice demonstrated clear indications of sleep disturbance in the form of decreased NREM sleep durations and bout counts, which declined to a greater extent by 10 months. Furthermore, a decrease in the REM sleep duration, bout counts, and theta band peak frequency, along with extended bout lengths occurred at 10 months in the 5XFAD mice compared to the control mice. Neuronal oscillations are believed to serve critical roles in cognitive function and information processing [23]. Numerous studies have identified relationships between neurodegenerative diseases and altered oscillatory activity during sleep and wakefulness [24,25,26]. REM theta oscillations, which are regulated by the medial septum, are a hallmark of REM sleep efficiency and have been shown to be associated with memory consolidation [27]. While the REM theta power in the 5XFAD mice did not decrease compared to the control mice, the decline in the theta band peak frequency over time indicates a potential REM sleep inefficiency. Peak theta frequency during REM sleep indicates a slowing of neuronal firing at the medial septum. The suppression of the cholinergic neurons of the LDT, a primary REM sleep-promoting neuronal group, may result in the reduced activation of the medial septum during REM sleep, which will consequently slow its firing rate and thereby interfere with memory consolidation. Interestingly, the sleep disturbances of 5XFAD mice were more prominent during the dark (active) phase, which suggests hyperactive behavior that is reminiscent of the Sundowning phenomenon, in which individuals with AD demonstrate elevated activity or agitation during the late afternoon or early morning [28]. Our data showed that NREM sleep decreased from 6 months in the 5XFAD mice compared to the controls, whereas decreased REM was observed only at 10 months. The progressing pattern of sleep disturbances are consistent with a human study of AD [4]. However, there are contradicting findings as well [24,29].

The neurocircuitry involved in sleep promotion is complex and encompasses neurons located in regions throughout the brain. Among these are the cortical nNOS neurons recognized for their roles in the generation of NREM sleep and NREM sleep homeostasis. Cortical nNOS neurons were the first neuronal population identified in the cerebral cortex with activation specific to slow wave sleep [30]. Because Aβ accumulation emerges near and within the cortex in early stages of AD, cortical nNOS neurons are likely among the first sleep-promoting neurons affected. As AD progresses, the amyloid pathology spreads toward the subcortical regions, including the hypothalamus and thalamus.

Galaninergic neurons of the VLPO are located in the anterior hypothalamus. These neurons are well known for their role in promoting NREM sleep and are suspected of generating NREM sleep through the inhibition of wake-promoting or wake-active neurons such as the locus coeruleus, tuberomammillary nucleus, and the dorsal raphe nucleus [31]. We anticipated that a significant disruption to the cortex and VLPO would result in declines in NREM sleep behavior, particularly NREM sleep durations. By 6 months, our data revealed reduced NREM sleep durations with increased Aβ levels, which implies a potential causal relationship between them. Because the number of nNOS neurons and galaninergic VLPO neurons remained unchanged, we postulated that the impact of Aβ on NREM sleep was mediated by Aβ pathology-related impairment of the sleep-promoting function of cortical nNOS neurons, galaninergic VLPO neurons, or both.

Reagrding changes in REM sleep behavior, we focused on the LDT. Cholinergic neurons of the LDT and pedunculopontine tegmentum promote REM sleep by depolarizing the neurons of the reticular formation [32], which play a role in the regulation of muscle tone to create muscle atonia [33]. The significance of cholinergic LDT neurons to REM sleep promotion has been exemplified in several studies, including one study that demonstrated increased REM sleep with optogenetic activation of the LDT [18] and another study that showed that the formation of lesions in the LDT resulted in reduced REM sleep [32]. In this study, REM sleep durations, bout counts, and theta peak frequencies did not decrease in the 5XFAD mice until 10 months when the amyloid pathology had increased significantly compared to the 3-month levels. This delay between the NREM sleep disturbance and REM sleep disturbance may result from changes in the spatiotemporal distribution of Aβ through the progression of AD [11,12,34]. Cortical nNOS neurons and galaninergic neurons of the VLPO would likely be affected by AD prior to cholinergic neurons of the LDT, which span the midbrain and pons. Counts of LDT neurons remained unchanged in the 5XFAD and control mice throughout all age groups. This indicates that changes in sleep behaviors did not arise from a loss of sleep-promoting neurons.

The oligomerization and fibrillization of Aβ and the formation of amyloid plaques in AD lead to the recruitment of astrocytes and their conversion into a reactive state, which potentially stimulates the progression of AD through astrocytic influences [35,36,37,38]. In addition to an elevated expression of GFAP, reactive astrocytes degrade Aβ and produce putrescine as a byproduct. Putrescine is further degraded by monoamine oxidase-b (MAO-B) to generate GABA [22]. The extrasynaptic release of GABA through bestrophin 1 channels of reactive astrocytes is believed to be capable of inhibiting nearby cells possessing extrasynaptic GABA receptors. Extrasynaptic GABA_A_ receptors are targets for various compounds including sleep-promoting drugs, alcohol, anesthetics, and more [39]. The δ subunit and the α5 subunit are components of extrasynaptic GABA_A_ receptors. We selected the δ subunit for this study because it is more commonly used in research to identify extrasynaptic GABA receptors than α5. Additionally, the δ subunit and the α5 subunit can cooccupy the same extrasynaptic GABA receptor. While there is a lower concentration of GABRD than synaptic GABA receptors, GABA has a higher affinity to GABRD than synaptic GABA receptors [39,40,41,42].

Previous research has demonstrated the aberrant and abundant release of GABA from reactive astrocytes to inhibit hippocampal neurons, which resulted in memory impairment in a mouse model of AD [36]. Here, astrocytes were stained for GFAP and GABA as markers to indicate the severity of astrogliosis. GABA is an inhibitory neurotransmitter and may have an inhibitory effect on neighboring cells containing extrasynaptic GABA receptors. Existing research yields conflicting findings regarding the presence of GABRD in specific neuronal groups. While several studies validate the existence of the GABRD in the neocortex, reports on their presence in the hypothalamus, pons, and midbrain are less consistent [40,41,42]. Here, GABRD staining of nNOS neurons in the cortex, galaninergic neurons of the VLPO and cholinergic neurons of the LDT confirmed the presence of extrasynaptic GABA_A_ receptors in each group, which suggests their susceptibility to inhibition by reactive astrocytic GABA.

The notion of an inhibitory role of reactive astrocytic GABA on sleep disturbance was supported by a correlation analysis using pooled data of the 5XFAD and control mice. The number of GABA-positive astrocytes at sleep-promoting areas in the 5XFAD mice increased along with plaque and astrocyte counts through the progression of AD. The proportions of neurotoxic reactive astrogliosis in the cortex and VLPO were negatively correlated with NREM sleep duration and NREM bout counts and positively correlated with NREM bout lengths. Additionally, the proportions of neurotoxic reactive astrogliosis in the LDT were negatively correlated with REM sleep duration, REM bout counts, and theta band peak frequency, and positively correlated with REM bout lengths.

Furthermore, decreases in the NREM sleep duration and bout counts appeared at the same age (6 months) that neurotoxic reactive astrogliosis increased in the cortex and VLPO in the 5XFAD mice compared to the control mice. However, a significant increase in the total astrocytes in the VLPO was not observed until 10 months in the 5XFAD mice compared to the controls. Similarly, decreases in the REM sleep duration and bout counts appeared at the same age (10 months) that neurotoxic reactive astrogliosis increased in the LDT in the 5XFAD mice compared to the control mice. However, total astrocyte counts in the LDT increased as early as 6 months in the 5XFAD mice compared to the controls. Although the total astrocyte counts also correlated with many parameters of sleep disturbance, the age differences between the initial increases in the total astrocyte counts between the 5XFAD and control mice and the emergence of sleep disruptions may suggest that astrocytic abundance alone is not the cause of sleep disruption. Rather, the correlations between neurotoxic reactive astrogliosis and changes in sleep behavior combined with the coinciding emergence of neurotoxic reactive astrogliosis and sleep disturbance may imply that sleep disturbance is mainly a consequence of GABA-positive astrogliosis. However, these relationships would need to be confirmed by additional research before causation can be concluded.

Another hypothesis claims that sleep disturbances and neuroinflammation have a bidirectional relationship in AD that involves the excessive release of inflammatory cytokines from microglia. While inflammation may serve a beneficial role of removing cellular debris, chronic inflammation can disrupt many cellular functions and ultimately impair sleep–wake cycles [43,44]. The astrocyte-regulated waste clearance process known as the glymphatic system operates primarily during sleep. Therefore, impairments to the sleep–wake cycle may results in a positive feedback loop of increasing Aβ levels, inflammation, and sleep disturbances [45]. Interestingly, the release of astrocytic GABA has been also demonstrated to inhibit microglial inflammatory responses and suppress proinflammatory signaling [46]. Therefore, there might be a beneficial function of astrocytic GABA as a response mechanism to counteract excessive inflammation.

A close examination of the sleep bout characteristics and EEG power spectra yielded a couple of surprising findings. First, it was originally unexpected to observe increases in the REM bout length in the aged 5XFAD mice compared to the controls. A progressive amyloid pathology intuitively led to reduced REM sleep, which we suspected would arise from a decrease in the REM bout lengths. However, our observations of longer REM bout lengths in the aged 5XFAD mice may be evidence of a REM rebound effect. NREM sleep precedes REM sleep in the natural sleep cycle. Therefore, as the proportion of time spent in NREM decreases, the opportunities to enter REM sleep are diminished. The 5XFAD mice may compensate for lost REM sleep opportunities by extending the duration of time spent in bouts of REM sleep. Sleep deprivation has been shown to lead to a REM sleep rebound in the form of longer REM sleep episodes and decreased NREM sleep intensity [47].

Second, we originally expected that the NREM delta power would decrease with the progression of AD, particularly as the nNOS neurons became progressively exposed to GABA. The NREM delta power is an indicator of sleep depth, and deep sleep is generally associated with a healthy NREM sleep efficiency. A previous study showed that the deletion of nNOS neurons in nNOS knockout mice decreased NREM sleep proportions, shortened NREM bout durations, and diminished the NREM delta power between 0.5 and 2.5 Hz [17]. However, unlike in the nNOS knockout mice, the AD pathology in 5XFAD mice did not result in the loss of nNOS neurons. Therefore, nNOS neurons likely maintained at least a partial homeostatic function. This may explain why the reduction in the NREM delta power was one of the earliest changes in sleep behavior observed in the 5XFAD mice compared to the control mice and why the reduction was not permanent. At 3 months, the AD pathology in the cortex may have been sufficient to disturb the nNOS neuronal regulation of the homeostatic sleep drive but insufficient to reduce NREM sleep durations. However, the decrease in the NREM sleep duration and bout counts over time potentially resulted in elevated sleep pressure as the Aβ pathology accelerated. The increase in the NREM delta power with age observed in the 5XFAD mice may have been a homeostatic regulatory mechanism compensating for the loss of NREM sleep duration. A homeostatic response to the decreased NREM sleep duration may also explain the positive correlation between the GABA-positive astrocytes and NREM bout length.

This study contains several potential limitations including the limited number of sleep-promoting brain regions investigated and the lack of an assessment on the wake-promoting neurons. Because the neurocircuitry of sleep regulation is not yet fully understood, we are unable to determine the extent that sleep was influenced by other sleep-regulating regions that were not investigated during this study. It is possible that certain wake-promoting neuron groups possess GABRD and, if impacted by amyloidosis or astrogliopathy, they may promote sleep and have important implications on our understanding of the hierarchy within sleep–wake neurocircuitry.

Another limitation involves the use of the 5XFAD mouse model. We acknowledge that the 5XFAD model primarily represents familial AD. While this model is useful in our investigation of the Aβ pathology’s impact on sleep disturbances, it does not fully capture the broad spectrum of AD cases, particularly sporadic AD. Our findings should be interpreted within the context of Aβ-related pathology and reactive astrogliosis. Furthermore, the natural sleeping behavior of mice differs from humans. Mice exhibit polyphasic sleep characteristics with approximately two-thirds of their sleep occurring during light and one-third occurring during darkness in healthy subjects [48]. Mice may experience up to 140 sleep cycles in a 24 h period [48]. This is in contrast to monophasic humans, in which healthy individuals typically experience between 1–8 cycles per night [48]. These differences must be taken in context when interpreting the changes in sleep behavior, particularly for the altered bout lengths and counts observed in mice. There are additional studies that analyzed sleep behavior in 5XFAD mice that yielded results that differ from this study [1,49,50]. However, the discrepancies were likely due to differences in the age and sex of the mice, as well as different experimental techniques.

Future studies incorporating experimental techniques such as the patch clamp may be useful in validating neuronal inhibition as the functional change occurring in sleep-promoting regions in 5XFAD mice with sleep deterioration. Furthermore, treatment with selective MAO-B inhibitors such as KDS2010 may be effective at restoring sleep in 5XFAD mice and can thereby confirm astrocytic GABA as the pathogenic mechanism causing sleep disturbances in AD.

## 4. Materials and Methods

### 4.1. Animals

Male 5XFAD mice and wild-type littermates were used for the investigation and were divided into 3-, 6-, and 10-month-old age groups (6 groups in total, n = 10 to 12 mice per group; Tg6799, Jackson Laboratory). 5XFAD mice are a double transgenic model of AD that overexpresses the amyloid precursor protein (APP) with Swedish (K670N, M671L), Florida (I716V), and London (V717I) mutations, as well as human PS1 with M146L and L286V mutations under the control of the murine Thy-1-promotor. The animals were housed in a temperature-controlled room at 23 ± 2 °C with a 12/12 h light/dark cycle. Food and water were provided ad libitum. All the mouse experiments were conducted according to the ethical guidelines of the Institutional Animal Care and Use Committee (GIST-2020-031).

### 4.2. Surgery

We implanted screw electrodes on the mouse skull (frontal: AP = 1.0 mm, ML = 1.0 mm from bregma; parietal AP = 1.0 mm, ML = 1.0 mm from lambda) while the mice were under isoflurane anesthesia (1–3%, Hana Pharm. Co., Ltd., Gyeonggi-do, Seoul, Republic of Korea). Electrodes for electromyography (EMG) were placed in the neck muscle. The mice were allowed to be in their home cages for 7 days for recovery with appropriate monitoring.

### 4.3. Sleep Recording and Analysis

After recovery, the mice were habituated to a recording chamber and were connected to the EEG recording system for 24 h. An EEG, EMG, and video were recorded for 24 h in freely moving conditions (8200-K1-SL, Pinnacle Technology, Lawrence, KS, USA). The data were divided into 10 s epochs and were analyzed for sleep–wake behavior using commercial software (Sirenia Sleep Pro 2.2.1 software, Pinnacle Technology, Lawrence, KS, USA). Briefly, wakefulness was identified by low-amplitude EEG oscillations and elevated EMG activity with phasic bursts; NREM sleep by low-frequency high-amplitude EEG oscillations with reduced EMG activity; and REM sleep by prominent theta rhythms with an absence of EMG activity. The following parameters were calculated: proportions of wake, NREM, REM, total sleep (TS = NREM + REM), and REM/TS ratio. In addition, lengths and counts of the NREM and REM sleep and wake bouts were also evaluated. Bouts were defined as three consecutive epochs with the same state. The average absolute power values were calculated for the following frequency bands: delta (0.5–4 Hz) and theta (4–8 Hz).

### 4.4. Immunohistochemistry

The paraformaldehyde-fixed brain tissues of the left hemisphere were cut into coronal sections (40 μm) with a cryostat (Leica Biosystems, Buffalo Grove, IL, USA). The sections were washed in 1% PBST and incubated for 2 h in a blocking solution (0.5% Triton X-100, 3% donkey serum in 0.1M PBS) after undergoing shaking at 100 rpm at room temperature. Primary antibodies were diluted in a blocking buffer (3% normal donkey serum 0.5% PBST). A complete list of antibody details is provided in Appendix A. The brain tissues with primary antibodies were incubated overnight at room temperature. After washing in PBS 3 times, the tissues were incubated for 2 h at room temperature with secondary antibodies. Aβ plaques were stained with Thioflavin S (1 mM, T1892, Sigma Aldrich, St. Louis, MO, USA). The tissues were mounted onto a saline-coated slide glass (5116-20F, Muto Pure Chemicals, Tokyo, Japan). A total of eight brain slices were used for a histological analysis: six from the cortical area (AP: +2.0, +1.0, 0, −1.0, −2.0, and −3.0 mm from bregma), one from VLPO (AP: 0 mm from bregma), and one from LDT (AP: −5.0 mm from bregma). Histological images were acquired with a confocal microscope (FV3000RS, Olympus, Tokyo, Japan), and Z-stacks of 20 μm in 2 μm steps were captured by using Olympus Fluoview software 2.5.1.228 (FV31S-SW, Olympus, Tokyo, Japan) with a 40× objective. We used the ‘Analyze Particles’ tool of the ImageJ software 1.53t (National Institutes of Health, Bethesda, MD, USA) to quantify the plaques and cells. Additionally, GABA-positive astrocytes were analyzed by quantifying the colabeling of GFAP and GABA by using the ‘Colocalization Threshold’ plugin in ImageJ software. The anatomical identification of the regions of interest in each coronal section was based on the mouse brain atlas of Paxinos and Franklin [51].

### 4.5. Statistical Analysis

The statistical analysis was performed by using the Origin software package 2021.9.8.0.200 (Northampton, MA, USA) and Prism 9 (GraphPad Software, Inc., La Jolla, CA, USA). All data are presented as mean ± SEM. Normality was determined by a Shapiro-Wilk test. Comparisons of 2 groups were carried out with an independent sample *t*-test or Mann–Whitney test. Comparisons among three groups were carried out with a one-way or two-way AVONA with a post hoc Tukey test or Kruskal–Wallis test. A value of *p* < 0.05 was considered statistically significant.

## 5. Conclusions

We demonstrated that 5XFAD mice exhibited sleep disturbances related to the distribution and severity of the Aβ pathology, particularly at sleep-promoting regions of the brain. Interestingly, changes in sleep behavior occurred despite sleep-promoting neuron counts remaining consistent. Rather, these changes primarily occurred in a progressive manner that coincided with the increased amyloid accumulation and correlated with the GABA-positive reactive astrogliosis at specific brain regions associated with NREM and REM sleep promotion and sleep homeostasis. Sleep is required for the activation of brain waste clearance processes such as the glymphatic system. Therefore, targeting sleep disturbances may offer several therapeutic benefits towards reducing the risks associated with AD.

## Figures and Tables

**Figure 1 ijms-24-09695-f001:**
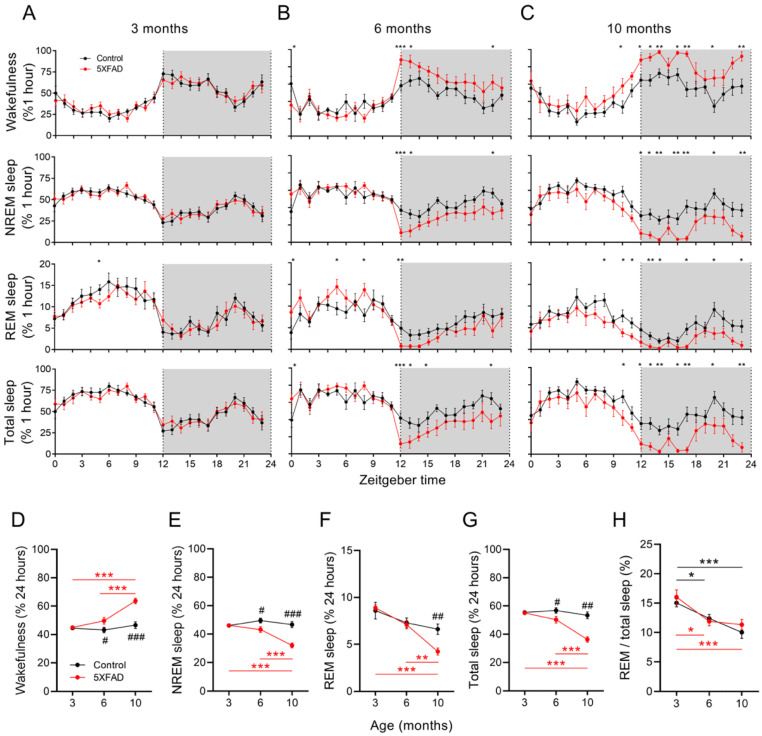
Diminished NREM and REM sleep duration with age in 5XFAD mice. (**A**–**C**) The 24 h time course for each vigilant state and total sleep in control and 5XFAD mice at 3, 6, and 10 months of age. (**D**–**H**) Time spent in wakefulness, NREM, REM, total sleep, and REM proportion for control and 5XFAD mice at 3, 6, and 10 months of age. Data are presented as mean ± SEM (n = 10–12 mice per group). * *p* < 0.05, ** *p* < 0.01, and *** *p* < 0.001 in red (5XFAD) and black (control) by one-way ANOVA or Kruskal–Wallis among age groups (all panels). ^#^
*p* < 0.05, ^##^
*p* < 0.01, ^###^
*p* < 0.001 by independent *t*-test or Mann–Whitney u test between genotypes (panels (**D**–**H**)). Abbreviations: ANOVA, analysis of variance; NREM, nonrapid eye movement; REM, rapid eye movement.

**Figure 2 ijms-24-09695-f002:**
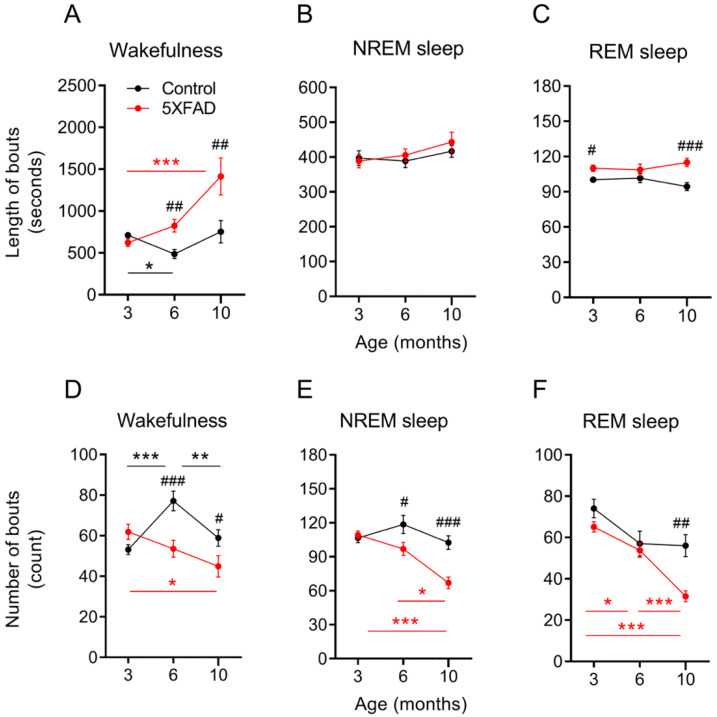
Bout counts of NREM and REM sleep reduced with age in 5XFAD mice. (**A**–**C**) Bout lengths during wakefulness, NREM, and REM sleep in control and 5XFAD mice at 3, 6, and 10 months of age. (**D**–**F**) Bout counts of 24 h EEG recording session for wakefulness, NREM, and REM sleep for control and 5XFAD mice at 3, 6, and 10 months of age. Data are presented as mean ± SEM (n = 10–12 mice per group). * *p* < 0.05, ** *p* < 0.01, and *** *p* < 0.001 in red (5XFAD) and black (control) by one-way ANOVA or Kruskal–Wallis among age groups (all panels). ^#^
*p* < 0.05, ^##^
*p* < 0.01, ^###^
*p* < 0.001 by independent *t*-test or Mann–Whitney u test between genotypes (all panels). Abbreviations: ANOVA, analysis of variance; EEG, electroencephalogram; NREM, nonrapid eye movement; REM, rapid eye movement.

**Figure 3 ijms-24-09695-f003:**
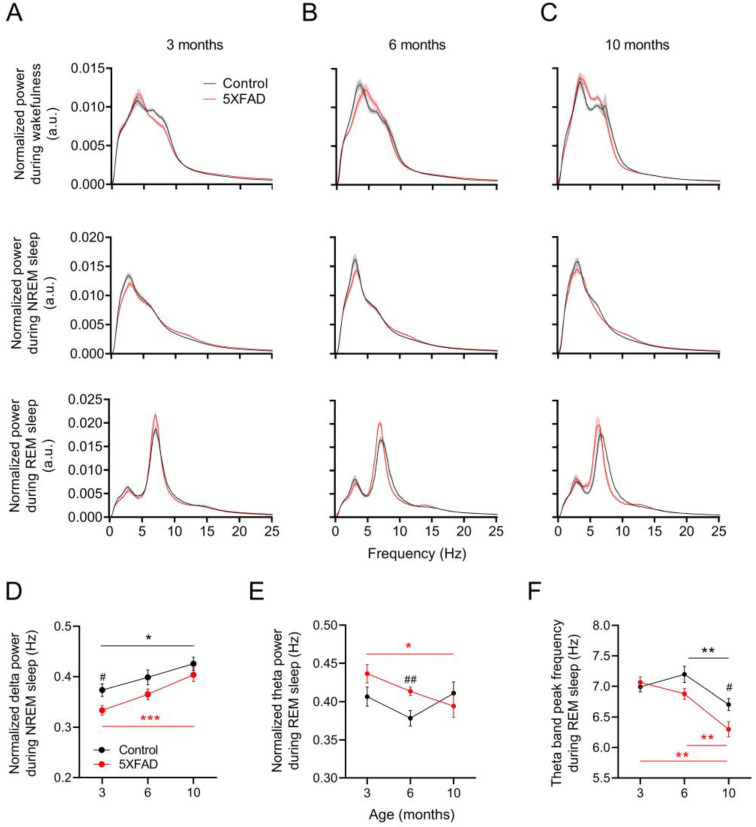
Changes in EEG power spectrum with age in 5XFAD mice. (**A**–**C**) Averaged power spectra of 10 s epochs of wakefulness, NREM, and REM sleep in control and 5XFAD mice at 3, 6, and 10 months of age. (**D**) Normalized delta power during NREM sleep in control and 5XFAD mice at all ages. (**E**) Normalized theta power during REM sleep in control and 5XFAD mice at all ages. (**F**) Peak frequency of theta band during REM sleep in control and 5XFAD mice at all age groups. Data are presented as mean ± SEM (n = 10–12 mice per group). * *p* < 0.05, ** *p* < 0.01, and *** *p* < 0.001 by one-way ANOVA or Kruskal–Wallis among age groups for control (black) and 5XFAD (red) mice (panels (**D**–**F**)). ^#^
*p* < 0.05, ^##^
*p* < 0.01 by independent *t*-test or Mann–Whitney u test between genotypes (panels (**D**–**F**)). Abbreviations: ANOVA, analysis of variance; EEG, electroencephalogram; NREM, nonrapid eye movement; REM, rapid eye movement.

**Figure 4 ijms-24-09695-f004:**
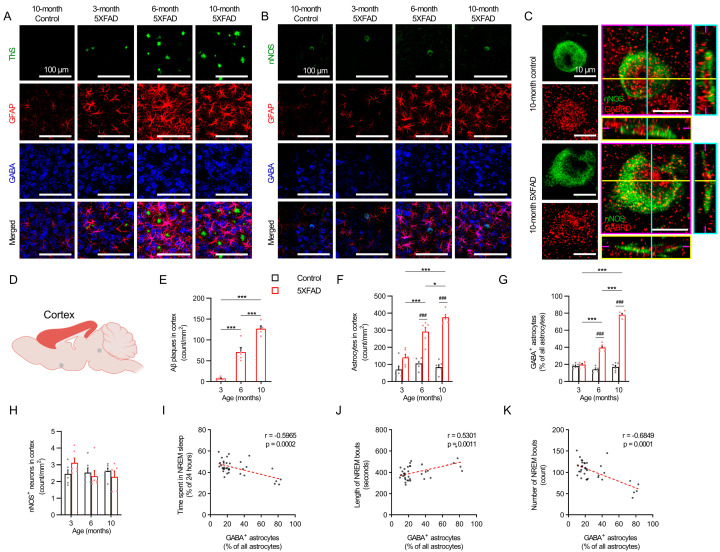
GABA-expressing astrocytes, cortical nNOS neurons, and the corresponding NREM sleep trends in 5XFAD mice. (**A**) Representative images of GABA-positive reactive astrocytes near the Aβ plaques in primary somatosensory cortex. Scale bar indicates 100 μm. (**B**) Representative images of GABA-positive reactive astrocytes near the nNOS neurons in primary somatosensory cortex. Scale bar indicates 100 μm. (**C**) Representative images of GABA_A_ δ subunit expression in the cortical nNOS neurons. Magenta, yellow, and cyan lines and image borders represent the orthogonal planes of the x-y, x-z, and y-z axes, respectively. Scale bar indicates 10 μm. (**D**) Schematic of cortical region examined. (**E**) The number of Aβ plaques per area unit in cortical area. Red bars indicate 3-, 6-, and 10-month-old 5XFAD mice. (**F**) The number of cortical astrocytes per area unit. (**G**) GABA-positive reactive astrocytes ratio in cortical area. (**H**) The number of cortical nNOS neurons per area unit. (**I**) The correlation between the ratio of cortical GABA-positive reactive astrocytes and NREM recording time (%). (**J**) The correlation between cortical GABA-positive reactive astrocytes ratio and NREM bout length. (**K**) The correlation between cortical GABA-positive reactive astrocytes ratio and NREM bouts counts. Black and red bars indicated the mean value and raw data in control and 5XFAD mice, respectively. Each dot represents the raw data. Data are presented as mean ± SEM (n = 5–6 mice per group). * *p* < 0.05, *** *p* < 0.001 between ages, ^###^
*p* < 0.001 between genotypes by one-way ANOVA with Tukey post hoc test (panel (**E**)), two-way ANOVA test with Tukey post hoc test (panels (**F**–**H**)), Pearson correlation test (panels (**I**–**K**)). Abbreviations: ANOVA, analysis of variance; EEG, electroencephalogram; GABA, gamma aminobutyric acid; GFAP, glial fibrillary acidic protein; nNOS, neuronal nitric oxide synthase; NREM, nonrapid eye movement; REM, rapid eye movement.

**Figure 5 ijms-24-09695-f005:**
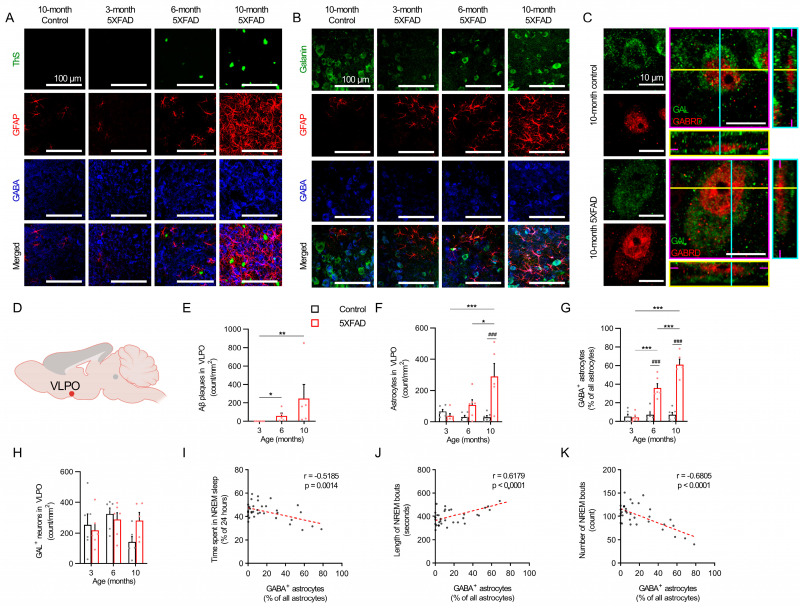
GABA-expressing astrocytes, galaninergic VLPO neurons, and the corresponding NREM sleep trends in 5XFAD mice. (**A**) Representative images of GABA-positive reactive astrocytes near the Aβ plaques in VLPO region. Scale bar indicates 100 μm. (**B**) Representative images of GABA-positive reactive astrocytes near the galaninergic neurons in VLPO region. Scale bar indicates 100 μm. (**C**) Representative images of GABA_A_ δ subunit expression in the VLPO galaninergic neurons. Magenta, yellow, and cyan lines and image borders represent the orthogonal planes of the x-y, x-z, and y-z axes, respectively. Scale bar indicates 10 μm. (**D**) Schematic of VLPO region examined. (**E**) The number of Aβ plaques per area unit in VLPO region. Red bars indicate 3-, 6-, and 10-month-old 5XFAD mice. (**F**) The number of astrocytes in the VLPO region per area unit. (**G**) GABA-positive reactive astrocytes ratio in cortical area. (**H**) The number of galaninergic neurons in the VLPO region per area unit. (**I**) The correlation between the ratio of GABA-positive reactive astrocytes in the VLPO region and NREM recording time (%). (**J**) The correlation between cortical GABA-positive reactive astrocytes ratio and NREM bout length. (**K**) The correlation between cortical GABA-positive reactive astrocytes ratio and NREM bouts counts. Black and red bars indicate the mean value and raw data in control and 5XFAD mice, respectively. Each dot represents the raw data. Data are presented as mean ± SEM (n = 5–6 mice per group). * *p* < 0.05, ** *p* < 0.01, *** *p* < 0.001 between ages, ^###^
*p* < 0.001 between genotypes by Kruskal–Wallis test with Dunn’s post hoc test (panel (**E**)), two-way ANOVA test with Tukey post hoc test (panels (**F**–**H**)), Pearson correlation test (panels (**I**–**K**)). Abbreviations: ANOVA, analysis of variance; GABA, gamma aminobutyric acid; GFAP, glial fibrillary acidic protein; NREM, nonrapid eye movement; VLPO, ventrolateral preoptic area.

**Figure 6 ijms-24-09695-f006:**
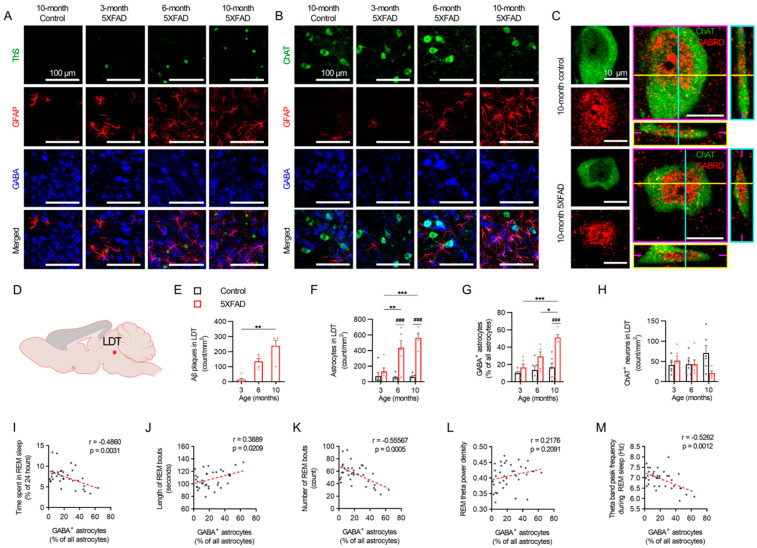
GABA-expressing astrocytes, cholinergic LDT neurons, and the corresponding REM sleep trends in 5XFAD mice. (**A**) Representative images of GABA-positive reactive astrocytes near the Aβ plaques in LDT region. Scale bar indicates 100 μm. (**B**) Representative images of GABA-positive reactive astrocytes near the cholinergic neurons in LDT region. Scale bar indicates 100 μm. (**C**) Representative images of GABA_A_ δ subunit expression in the cholinergic neurons at LDT region. Magenta, yellow, and cyan lines and image borders represent the orthogonal planes of the x-y, x-z, and y-z axes, respectively. Scale bar indicates 10 μm. (**D**) Schematic of LDT region examined. (**E**) The number of Aβ plaques per area unit in LDT region. (**F**) The number of astrocytes in LDT region per area unit. (**G**) GABA-positive reactive astrocytes ratio in LDT region. (**H**) The number of cholinergic neurons in LDT region per area unit. (**I**–**M**) The correlation between GABA-positive reactive astrocytes ratio at LDT region and REM duration, bout length, bout count, theta power, and theta peak frequency. Black and red bars indicate the mean value and raw data, respectively, in control and 5XFAD mice. Each dot represents the raw data. Data are presented as mean ± SEM (n = 5–6 mice per group). * *p* < 0.05, ** *p* < 0.01, *** *p* < 0.001 between ages, ^###^
*p* < 0.001 between genotypes by one-way ANOVA test with Tukey post hoc test (panel (**E**)), two-way ANOVA test with Tukey post hoc test (panels (**F**–**H**)), and Pearson correlation test (panels (**I**–**M**)). Abbreviations: ANOVA, analysis of variance; ChAT, choline acetyltransferase; EEG, electro-encephalogram; GABA, gamma aminobutyric acid; GFAP, glial fibrillary acidic protein; LDT, laterodorsal tegmentum; NREM, nonrapid eye movement; REM, rapid eye movement.

## Data Availability

Not applicable.

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
