# Peer review of "GABA-Positive Astrogliosis in Sleep-Promoting Areas Associated with Sleep Disturbance in 5XFAD Mice"

_ijms, 2023, doi:10.3390/ijms24119695_

Round 1
Reviewer 1 Report
This article did a good job of recording sleep changes in the 5xFAD model mice at different ages and suggesting possible mechanisms for the occurrence of sleep disorders in the AD model mice, but there are several issues to be aware of:
1. The background section mentioned that the proportion of AD patients with sleep disorders in the clinic was one-third, so did all of the 5xFAD model mice have sleep disorders in the study? Is it plausible (or is it consistent with the phenomenon of clinical research?) to use this model mouse for studies on sleep 2;
2. Determination of GABA-expressing astrocytes: Is it rigorous to determine GABA-expressing astrocytes in this study using the criteria of GABA and GFAP co-labeling in immunohistochemistry? First, the authors should stain the neurons to distinguish whether the released GABA is disturbed by GABAergic neurons, e.g. whether the number of GABAergic neurons or the proportion of GABA released is altered over the pathological process. Furthermore, it is not rigorous to show that GABA released by astrocyte can act on neurons only by GABRD labelling of postsynaptic membrane GABA receptors. Although the deficiency of not using electrophysiological validation has been described in the Discussion section, this may be an important support for the innovation aspect of this study. Could the validation of GABA-expressing astrocytes be aided by the use of some molecular markers? Determination of the bestrophin 1 and GABA-permeable Swell1 channels in these three regions (Junhua Yang et al., Neuron, 2023, https://doi.org/10.1016/j.neuron.2022.12.033) might provide the evidence for this study. in the absence of electrophysiological conditions.
3. There are some problems with the layout of the images: some of the images in Figure 3 are not aligned; the layout of the images in Figures 4 and 5 could be spruced up a bit.
Reviewer 2 Report
In the manuscript submitted by Drew et.al. titled as ‘GABA-Positive Astrocytosis in Sleep-Promoting Areas Associated with Sleep Disturbance in 5XFAD Mice’, The authors assessed the EEG profiles of 5xFAD mice and control mice at different ages, and examined the activity of astrocytes in sleep regulating brain regions through the immunostaining of GFAP and GABA. Their results revealed negative correlation of the reduced NREM or REM with the increased reactivity of astrocytes.
Overall, this paper is of significance to novelty to understand the relation of glial pathologies with the disturbance of sleep in AD mouse models. The paper itself is well written and organized with clear logic. Still, I have some comments below.Major
1. The quantification of the immunostaining figures were ommitted. Please clarify the imaging information, how many fields per sample? Whether z-stack were used? 20x or 40x? Please clarify how the authors measure the plaque number and cell numbers in ImageJ, through certain plugins or mannually? Specifically inFig 4, please clarify which cortex regions were images and applied in the analysis. Which layer? Or certain cortex region?
2. Line 95-97, Line 122-123, Line 154-156, Line 210-211, Line230-231, the statistical methods are not specific to specific panel. Please clarify this issue. Also in figure 1-3, please state clearly if it is one-way ANOVA.
Minor
1. Line 474, please clarify which 5xFAD line was used, 034840-JAX, or 034848-JAX?
2. Line 83, please clarify the meaning of ‘TS’ in the ‘REM/TS ratio’. Does it mean ‘total sleep’ time?
3. Could the authors elaborate the biological meaning on the reduced peak frequency in the 5xFAD mice in Fig. 3F?
4. Please clarify the inconsistence in the definition of the theta frequencies in Line 500 and Line 130.
5. We usually use astrogliosis instead of Astrocytosis.
Author Response
Reviewer 2
Re Major Comment #1
The quantification of the immunostaining figures were ommitted. Please clarify the imaging information, how many fields per sample? Whether z-stack were used? 20x or 40x? Please clarify how the authors measure the plaque number and cell numbers in ImageJ, through certain plugins or mannually? Specifically inFig 4, please clarify which cortex regions were images and applied in the analysis. Which layer? Or certain cortex region?
Ans) Apologies for the insufficient information. For the cortical nNOS neuron analysis, we used 6 brain slices (AP: +2.0, +1.0, 0, -1.0, -2.0, -3.0 mm from bregma) in one mouse brain. Because sleep-promoting nNOS neuron is widespread in the cortex, we analyzed the broad cortical areas. In Fig. 4, representative images were obtained in primary somatosensory cortex layer V and VI. The other brain region, one brain slice in one mouse brain was used in VLPO (AP: 0 mm from bregma) and LDT (AP: -5.0 mm from bregma). The images were obtained with 40x objective, and 20-μm Z-stack images in 2-μm steps. To measure the number of plaques and cell, we used the ‘Analyze Particle’ tool in ImageJ software. Also, to measure the GABA-positive astrocytes, we used the ‘Colocalization Threshold’ plugin in ImageJ software to analyze colocalization between astrocytes and GABA. We described this information in the Figure 4 legend and Method and Material part.
Original Figure Legend:
Figure 4. … (A) Representative images of GABA-positive reactive astrocytes near the Aβ plaques in cortical area. Scale bar indicates 100 μm. (B) Representative images of GABA-positive reactive astrocytes near the cholinergic neurons in cortical area. Scale bar indicates 100 μm. … *p < 0.05, **p < 0.01, ***p < 0.001 by Two-way ANOVA test with Tukey post hoc method, Kruskal-Wallis test or Pearson correlation test.
Revised Figure Legend:
Figure 4. … (A) Representative images of GABA-positive reactive astrocytes near the Aβ plaques in primary somatosensory cortex. Scale bar indicates 100 μm. (B) Representative images of GABA-positive reactive astrocytes near the nNOS neurons in primary somatosensory cortex. Scale bar indicates 100 μm. … *p < 0.05, **p < 0.01, ***p < 0.001 by one-way ANOVA with Tukey post hoc test (panel E), two-way ANOVA test with Tukey post hoc test (panels F-H), Pearson correlation test (panels I-K).
Original text:
4.4. Immunohistochemistry
The paraformaldehyde-fixed brain tissues of the left hemisphere were cut into coronal sections (40 μm) with a cryostat (Leica Biosystems, Buffalo Grove, IL). Sections were washed in 1% PBST and incubated for 2 hours in a blocking solution (0.5% Triton X-100, 3% donkey serum in 0.1M PBS) shaking at 100 rpm at room temperature. Primary antibodies were diluted in blocking buffer (3% normal donkey serum 0.5% PBST). A complete list of antibody details is provided in Supplementary Table S1. The brain tissues with primary antibodies were incubated overnight at room temperature. After washing in PBS 3 times, tissues were incubated for 2 hours at room temperature with secondary antibodies. Aβ plaques were stained with Thioflavin S (1 mM, T1892, Sigma Aldrich, St Louis, MO). The tissues were mounted onto a silane-coated slide glass (5116-20F, Muto Pure Chemicals, Tokyo, Japan). Histological images were acquired with a confocal microscope (FV3000RS, Olympus, Tokyo, Japan) and analyzed using ImageJ software (National Institutes of Health, Bethesda, MD). The mouse brain areas were identified based on Paxinos and Franklin's mouse brain atlas [33].
Revised text:
4.4. Immunohistochemistry
The paraformaldehyde-fixed brain tissues of the left hemisphere were cut into coronal sections (40 μm) with a cryostat (Leica Biosystems, Buffalo Grove, IL). Sections were washed in 1% PBST and incubated for 2 hours in a blocking solution (0.5% Triton X-100, 3% donkey serum in 0.1M PBS) shaking at 100 rpm at room temperature. Primary antibodies were diluted in blocking buffer (3% normal donkey serum 0.5% PBST). A complete list of antibody details is provided in Supplementary Table S1. The brain tissues with primary antibodies were incubated overnight at room temperature. After washing in PBS 3 times, tissues were incubated for 2 hours at room temperature with secondary antibodies. Aβ plaques were stained with Thioflavin S (1 mM, T1892, Sigma Aldrich, St Louis, MO). The tissues were mounted onto a silane-coated slide glass (5116-20F, Muto Pure Chemicals, Tokyo, Japan). A total of eight brain slices were used for histological analysis: six from the cortical area (AP: +2.0, +1.0, 0, -1.0, -2.0, and -3.0 mm from bregma), one from VLPO (AP: 0 mm from bregma), and one from LDT (AP: -5.0 mm from bregma). Histological images were acquired with a confocal microscope (FV3000RS, Olympus, Tokyo, Japan), and Z-stacks of 20 μm in 2 μm steps were captured using Olympus Fluoview software (FV31S-SW, Olympus, Tokyo, Japan) with 40x objective. We used the 'Analyze Particles' tool of the ImageJ software (National Institutes of Health, Bethesda, Maryland) for quantification of plaques and cells. Also, GABA-positive astrocytes were analyzed by quantifying the co-labeling of GFAP and GABA, using the 'Colocalization Threshold' plugin in ImageJ software. Anatomical identification of the regions of interest in each coronal section was based on the mouse brain atlas of Paxinos and Franklin [33].
Re Major Comment #2
Line 95-97, Line 122-123, Line 154-156, Line 210-211, Line230-231, the statistical methods are not specific to specific panel. Please clarify this issue. Also in figure 1-3, please state clearly if it is one-way ANOVA.
Ans) Thank you for your comment. We have corrected this issue.
Original Figure Legends:
Figure 1. … *p < 0.05, **p < 0.01, and ***p < 0.001 in red (5XFAD) and black (control) by ANOVA or Kruskal-Wallis among age groups (all panels). #p < 0.05, ##p < 0.01, ###p < 0.001 by independent t-test or Mann-Whitney u test between genotypes (panels D-H). Abbreviations: ANOVA, analysis of variance; EEG, electroencephalogram; NREM, non-rapid eye movement; REM, rapid eye movement.
Figure 2. … *p < 0.05, **p < 0.01, and ***p < 0.001 in red (5XFAD) and black (control) by ANOVA or Kruskal-Wallis among age groups. #p < 0.05, ##p < 0.01, ###p < 0.001 by independent t-test or Mann-Whitney u test between genotypes. Abbreviations: ANOVA, analysis of variance; EEG, electroencephalogram; NREM, non-rapid eye movement; REM, rapid eye movement.
Figure 3. … *p < 0.05, **p < 0.01, and ***p < 0.001 by ANOVA or Kruskal-Wallis among age groups for control (black) and 5XFAD (red) mice. #p < 0.05, ##p < 0.01, ###p < 0.001 by independent t-test or Mann-Whitney u test between genotypes. Abbreviations: ANOVA, analysis of variance; EEG, electroencephalogram; NREM, non-rapid eye movement; REM, rapid eye movement.
Figure 4. … (A) Representative images of GABA-positive reactive astrocytes near the Aβ plaques in cortical area. Scale bar indicates 100 μm. (B) Representative images of GABA-positive reactive astrocytes near the cholinergic neurons in cortical area. Scale bar indicates 100 μm. … *p < 0.05, **p < 0.01, ***p < 0.001 by Two-way ANOVA test with Tukey post hoc method, Kruskal-Wallis test or Pearson correlation test.
Figure 5. … (C) Representative images of GABA A δ subunit expression in the VLPO galaninergic neurons. Magenta, yellow and cyan lines and image borders represent the orthogonal planes of the x-y, x-z and y-z axes, respectively. Scale bar indicates 10 μm. ... *p < 0.05, **p < 0.01, ***p < 0.001 by Two-way ANOVA test with Tukey post hoc method, Kruskal-Wallis test or Pearson correlation test. Abbreviations: ANOVA, analysis of variance; EEG, electroencephalogram; GABA, gamma aminobutyric acid; GFAP, glial fibrillary acidic protein; NREM, non-rapid eye movement; REM, rapid eye movement; VLPO, ventrolateral preoptic area.
Figure 6. ... (C) Representative images of GABAA δ subunit expression in the cholinergic neurons at LDT region. Magenta, yellow and cyan lines and image borders represent the orthogonal planes of the x-y, x-z and y-z axes, respectively. Scale bar indicates 10 μm. ... *p < 0.05, **p < 0.01, ***p < 0.001 by Two-way ANOVA test with Tukey post hoc method, Kruskal-Wallis test or Pearson correlation test. Abbreviations: ANOVA, analysis of variance; ChAT, choline acetyltransferase; EEG, electro-encephalogram; GABA, gamma aminobutyric acid; GFAP, glial fibrillary acidic protein; LDT, laterodorsal tegmentum; NREM, non-rapid eye movement; REM, rapid eye movement.
Revised Figure Legends:
Figure 1. … *p < 0.05, **p < 0.01, and ***p < 0.001 in red (5XFAD) and black (control) by one-way ANOVA or Kruskal-Wallis among age groups (all panels). #p < 0.05, ##p < 0.01, ###p < 0.001 by independent t-test or Mann-Whitney u test between genotypes (panels D-H). Abbreviations: ANOVA, analysis of variance; EEG, electroencephalogram; NREM, non-rapid eye movement; REM, rapid eye movement.
Figure 2. … *p < 0.05, **p < 0.01, and ***p < 0.001 in red (5XFAD) and black (control) by one-way ANOVA or Kruskal-Wallis among age groups (all panels). #p < 0.05, ##p < 0.01, ###p < 0.001 by independent t-test or Mann-Whitney u test between genotypes (all panels). Abbreviations: ANOVA, analysis of variance; EEG, electroencephalogram; NREM, non-rapid eye movement; REM, rapid eye movement.
Figure 3. … *p < 0.05, **p < 0.01, and ***p < 0.001 by one-way ANOVA or Kruskal-Wallis among age groups for control (black) and 5XFAD (red) mice (panels D-F). #p < 0.05, ##p < 0.01, ###p < 0.001 by independent t-test or Mann-Whitney u test between genotypes (panels D-F). Abbreviations: ANOVA, analysis of variance; EEG, electroencephalogram; NREM, non-rapid eye movement; REM, rapid eye movement.
Figure 4. … (A) Representative images of GABA-positive reactive astrocytes near the Aβ plaques in primary somatosensory cortex. Scale bar indicates 100 μm. (B) Representative images of GABA-positive reactive astrocytes near the nNOS neurons in primary somatosensory cortex. Scale bar indicates 100 μm. … *p < 0.05, **p < 0.01, ***p < 0.001 by one-way ANOVA with Tukey post hoc test (panel E), two-way ANOVA test with Tukey post hoc test (panels F-H), Pearson correlation test (panels I-K).
Figure 5. ... (C) Representative images of GABAA δ subunit expression in the VLPO galaninergic neurons. Magenta, yellow and cyan lines and image borders represent the orthogonal planes of the x-y, x-z and y-z axes, respectively. Scale bar indicates 10 μm. ... *p < 0.05, **p < 0.01, ***p < 0.001 by Kruskal-Wallis test with Dunn’s post hoc test (panel E), two-way ANOVA test with Tukey post hoc test (panels F-H), Pearson correlation test (panels I-K). Abbreviations: ANOVA, analysis of variance; EEG, electroencephalogram; GABA, gamma aminobutyric acid; GFAP, glial fibrillary acidic protein; NREM, non-rapid eye movement; REM, rapid eye movement; VLPO, ventrolateral preoptic area.
Figure 6. … (C) Representative images of GABAA δ subunit expression in the cholinergic neurons at LDT region. Magenta, yellow and cyan lines and image borders represent the orthogonal planes of the x-y, x-z and y-z axes, respectively. Scale bar indicates 10 μm. ... *p < 0.05, **p < 0.01, ***p < 0.001 by one-way ANOVA test with Tukey post hoc test (panel E), two-way ANOVA test with Tukey post hoc test (panels F-H), Pearson correlation test (panels I-M). Abbreviations: ANOVA, analysis of variance; ChAT, choline acetyltransferase; EEG, electro-encephalogram; GABA, gamma aminobutyric acid; GFAP, glial fibrillary acidic protein; LDT, laterodorsal tegmentum; NREM, non-rapid eye movement; REM, rapid eye movement.
Re Minor Comment #1
Line 474, please clarify which 5xFAD line was used, 034840-JAX, or 034848-JAX?
Ans) We used 034840-JAX. This has been incorporated into the text.
Original Figure Legends:
Figure 1. Diminished NREM and REM sleep duration with age in 5XFAD mice. (A–C) The 24-hour time course for each vigilant state and total sleep in control and 5XFAD mice at 3, 6 and 10 months of age. (D–H) Time spent in wakefulness, NREM, REM, total sleep, and REM proportion for control and 5XFAD mice at 3, 6 and 10 months of age. Data were presented as mean ± SEM (n = 10 - 12 mice per group). *p < 0.05, **p < 0.01, and ***p < 0.001 in red (5XFAD) and black (control) by one-way ANOVA or Kruskal-Wallis among age groups. #p < 0.05, ##p < 0.01, ###p < 0.001 by independent t-test or Mann-Whitney u test between genotypes. Abbreviations: ANOVA, analysis of variance; EEG, electroencephalogram; NREM, non-rapid eye movement; REM, rapid eye movement.
Revised Figure Legends:
Figure 1. Diminished NREM and REM sleep duration with age in 5XFAD mice. (A–C) The 24-hour time course for each vigilant state and total sleep in control and 5XFAD mice at 3, 6 and 10 months of age. (D–H) Time spent in wakefulness, NREM, REM, total sleep, and REM proportion for control and 5XFAD mice at 3, 6 and 10 months of age. Data were presented as mean ± SEM (n = 10 - 12 mice per group). *p < 0.05, **p < 0.01, and ***p < 0.001 in red (5XFAD) and black (control) by one-way ANOVA or Kruskal-Wallis among age groups (all panels). #p < 0.05, ##p < 0.01, ###p < 0.001 by independent t-test or Mann-Whitney u test between genotypes (panels D-H). Abbreviations: ANOVA, analysis of variance; EEG, electroencephalogram; NREM, non-rapid eye movement; REM, rapid eye movement.
Re Minor Comment #2
Line 83, please clarify the meaning of ‘TS’ in the ‘REM/TS ratio’. Does it mean ‘total sleep’ time?
Ans) My apologies for the confusion. TS means total sleep. In REM/TS, I am referring to the proportion of the total sleep time spent in REM sleep.
Original text:
In longitudinal analyses, control mice showed no differences among all age groups in durations of vigilance states except in the REM/TS ratio (16.0% at 3 months and 11.3% at 10 months, p < 0.01; Figure 1H).
Revised text:
In longitudinal analyses, control mice showed no differences among all age groups in durations of vigilance states except in the REM/TS (REM sleep/total sleep) ratio (16.0% at 3 months and 11.3% at 10 months, p < 0.01; Figure 1H).
Re Minor Comment #3
Could the authors elaborate the biological meaning on the reduced peak frequency in the 5xFAD mice in Fig. 3F?
Ans) Peak theta frequency during REM sleep indicates a slowing of neuronal firing at the medial septum. The suppression of cholinergic neurons of the LDT, a primary REM sleep-promoting neuron, may result in reduced activation of the medial septum consequently slowing its firing rate. Theta oscillations of the medial septum have been linked to contextual memory performance (Boyce et al., 2016).
Original text:
Analysis of sleep-wake patterns in 5XFAD mice revealed a gradual decline of various sleep parameters with age. 5XFAD mice showed a similar sleep pattern with control mice at 3 months… REM theta is a hallmark of REM sleep efficiency and has been shown to be associated with memory consolidation [5]. While REM theta power in 5XFAD mice did not decrease compared to control mice, the decline in theta band peak frequency over time indicates a potential REM sleep inefficiency. Interestingly, the sleep disturbances of 5XFAD mice were more prominent during dark (active) phase, suggesting hyperactive behavior reminiscent of the Sundowning phenomenon, in which individuals with AD demonstrate elevated activity or agitation during late afternoon or early morning [45].
Revised text:
Analysis of sleep-wake patterns in 5XFAD mice revealed a gradual decline of various sleep parameters with age. 5XFAD mice showed a similar sleep pattern with control mice at 3 months… REM theta oscillations, regulated by the medial septum, are a hallmark of REM sleep efficiency and have been shown to be associated with memory consolidation [5]. While REM theta power in 5XFAD mice did not decrease compared to control mice, the decline in theta band peak frequency over time indicates a potential REM sleep inefficiency. Peak theta frequency during REM sleep indicates a slowing of neuronal firing at the medial septum. The suppression of cholinergic neurons of the LDT, a primary REM sleep-promoting neuronal group, may result in reduced activation of the medial septum during REM sleep consequently slowing its firing rate, thereby interfering with memory consolidation. Interestingly, the sleep disturbances of 5XFAD mice were more prominent during dark (active) phase, suggesting hyperactive behavior reminiscent of the Sundowning phenomenon, in which individuals with AD demonstrate elevated activity or agitation during late afternoon or early morning [45].
Reference:
Boyce, R., Glasgow, S.D., Williams, S., and Adamantidis, A. (2016). Causal evidence for the role of REM sleep theta rhythm in contextual memory consolidation. Science 352, 812-816.
Re Minor Comment #4
Please clarify the inconsistence in the definition of the theta frequencies in Line 500 and Line 130.
Ans) My apologies for the mistake. It has been corrected.
Re Minor Comment #5
We usually use astrogliosis instead of Astrocytosis.
Ans) I have changed astrocytosis to astrogliosis in throughout the manuscript.
Closing remarks:
I would like to express my gratitude for your time and effort in reviewing my work. Your valuable feedback is sincerely appreciated, as I believe it has significantly improved the quality of my work. Thank you once again.
Round 2
Reviewer 1 Report
The authors have answered all the questions, the manuscript can be accepted now.